# Anti-Fibrotic Effect of Human Wharton’s Jelly-Derived Mesenchymal Stem Cells on Skeletal Muscle Cells, Mediated by Secretion of MMP-1

**DOI:** 10.3390/ijms21176269

**Published:** 2020-08-29

**Authors:** Alee Choi, Sang Eon Park, Jang Bin Jeong, Suk-joo Choi, Soo-young Oh, Gyu Ha Ryu, Jeehun Lee, Hong Bae Jeon, Jong Wook Chang

**Affiliations:** 1Stem Cell Institute, ENCell Co. Ltd., Seoul 06072, Korea; ari0621@encellinc.com (A.C.); earnie.park@encellinc.com (S.E.P.); jb.jeong@encellinc.com (J.B.J.); 2Stem Cell & Regenerative Medicine Institute, Samsung Medical Center, Seoul 06351, Korea; 3Department of Obstetrics and Gynecology, Samsung Medical Center, Seoul 06351, Korea; drmaxmix.choi@samsung.com (S.-j.C.); ohsymd.oh@samsung.com (S.-y.O.); 4Department of Medical Device Management and Research, SAIHST, Sungkyunkwan University School of Medicine, Seoul 06351, Korea; gyuha.ryu@samsung.com; 5The Office of R&D Strategy & Planning, Samsung Medical Center, Seoul 06351, Korea; 6Department of Pediatrics, Samsung Medical Center, Seoul 06351, Korea; leejeehunmd@gmail.com

**Keywords:** Duchenne muscular dystrophy, matrix metalloproteinase-1, paracrine factor, skeletal muscle fibrosis, Wharton’s jelly-derived mesenchymal stem cell

## Abstract

Extracellular matrix (ECM) components play an important role in maintaining skeletal muscle function, but excessive accumulation of ECM components interferes with skeletal muscle regeneration after injury, eventually inducing fibrosis. Increased oxidative stress level caused by dystrophin deficiency is a key factor in fibrosis in Duchenne muscular dystrophy (DMD) patients. Mesenchymal stem cells (MSCs) are considered a promising therapeutic agent for various diseases involving fibrosis. In particular, the paracrine factors secreted by MSCs play an important role in the therapeutic effects of MSCs. In this study, we investigated the effects of MSCs on skeletal muscle fibrosis. In 2–5-month-old mdx mice intravenously injected with 1 × 10^5^ Wharton’s jelly (WJ)-derived MSCs (WJ-MSCs), fibrosis intensity and accumulation of calcium/necrotic fibers were significantly decreased. To elucidate the mechanism of this effect, we verified the effect of WJ-MSCs in a hydrogen peroxide-induced fibrosis myotubes model. In addition, we demonstrated that matrix metalloproteinase-1 (MMP-1), a paracrine factor, is critical for this anti-fibrotic effect of WJ-MSCs. These findings demonstrate that WJ-MSCs exert anti-fibrotic effects against skeletal muscle fibrosis, primarily via MMP-1, indicating a novel target for the treatment of muscle diseases, such as DMD.

## 1. Introduction

Skeletal muscle fibrosis, one of the typical symptoms of Duchenne muscular dystrophy (DMD), is an abnormal accumulation of extracellular matrix (ECM) components in muscle tissue and is particularly observed in patients with the end-stage of DMD [1,2]. In fibrotic skeletal muscle, the expression level of pro-fibrotic factors is increased, resulting in excessive deposition of ECM proteins, such as fibronectin. As a result, muscle function is impaired and the regeneration ability of skeletal muscle is diminished, which leads to muscle dysfunction [3,4,5]. Thus, this pathological symptom is considered a hallmark of muscular dystrophies.

Various studies have investigated the mechanisms underlying skeletal muscle fibrosis and have identified several factors that affect fibrosis, such as oxidative stress, inflammation, and aging [1,6,7]. In particular, oxidative stress in DMD, which is mainly caused by dystrophin deficiency, is considered to be one of the major causes of muscle fibrosis [8,9]. Therefore, alleviation of oxidative stress-induced fibrosis is important in relieving the pathologic condition in patients with DMD. There are several approaches to resolving fibrosis in patients with DMD, but no effective treatment has been developed to date [10]. 

Currently, numerous clinical trials have proven that mesenchymal stem cells (MSCs) are effective and safe for use in the treatment of various diseases [11,12,13,14]. Since MSCs derived from bone marrow, umbilical cord, or adipose tissues are adult stem cells, there are no safety or ethical issues involved [15,16]. Moreover, it is known that various proteins secreted from mesenchymal stem cells, i.e., paracrine factors, have therapeutic potentials [17,18,19]. The paracrine factors secreted by MSCs help to maintain homeostasis through the regulation of inflammation and the immune systems at the lesion site [20]. Recently, various paracrine factors have been investigated and related studies have suggested that these factors play a pivotal role in the therapeutic effects of MSCs [21,22,23]. It has been suggested that Wharton’s jelly (WJ)-MSCs, which exert anti-apoptotic, anti-inflammatory, and immunomodulatory effects by secreting paracrine factors, could be an effective therapeutic agent for muscle diseases [24,25,26]. We have previously demonstrated that WJ-MSCs exert anti-apoptotic effects on skeletal muscle cells under starvation condition, by the secretion of XCL1 protein [27]. However, to date, no study has investigated the effect of WJ-MSCs on fibrosis in patients with DMD. 

Therefore, in this study, we investigated whether MSCs attenuate skeletal muscle fibrosis in an mdx animal model. To figure out the mechanisms involved in anti-fibrotic effects of WJ-MSCs, we established a hydrogen peroxide-induced fibrosis model in an in-vitro system. Furthermore, we identified matrix metalloproteinase-1 (MMP-1) as the key paracrine factor that mediates anti-fibrotic effect.

## 2. Results

### 2.1. Anti-Fibrotic Effects of WJ-MSCs on an Mdx Model

The effects of WJ-MSCs on DMD pathology were evaluated in the gastrocnemius muscles of an mdx murine model. The level of fibrosis, presenting as the relative area stained with Sirius red solution, was increased in the mdx group as compared to the control group (Figure 1a, 514 ± 5% of the control group value). Additionally, accumulation of calcium/necrotic fibers was increased in the mdx group as compared to the control group (Figure 1b, 661 ± 68% of the control value).

In mdx mice intravenously (I.V) injected with WJ-MSCs, the severity of fibrosis and accumulation of calcium/necrotic fibers was decreased (Figure 1, 280 ± 5% of control value and 265 ± 28% of the control value, respectively). These data imply that WJ-MSCs have a positive effect on degenerative phenotypes of mdx mice.

### 2.2. Effects of WJ-MSCs on an In-Vitro H_2_O_2_-Induced Fibrosis Model

To investigate the role of WJ-MSCs on the fibrosis observed in mdx mice, we established an in-vitro skeletal muscle fibrosis model. There are many possible hypotheses about the cause of fibrosis in DMD, such as aging, trauma, or inflammation [4], but reactive oxygen species (ROS)-induced oxidative stress is considered as a major cause of this feature of DMD. To achieve oxidative stress-induced fibrosis, hydrogen peroxide (H_2_O_2_) of various concentrations were used to treat myotubes. We found that 2 mM H_2_O_2_ did not induce cell death of myotubes, but induced fibrosis phenotypes in the treated cells as compared to control cells (Figure A1 and Figure A2). Myotubes treated with 2 mM H_2_O_2_ showed damage (Figure 2a), and the area stained with Sirius red solution was increased as compared to the control group (Figure 2b, 133 ± 5% of control group value), indicating fibrotic degeneration. Similarly, the expression level of fibronectin protein, a major ECM protein, was increased in the treated group as compared to the control group (Figure 2c,d, 120 ± 5% of control group value). Along with this change, relative expression levels of myosin heavy chain (MHC), a myotube marker, was decreased in the treated group (Figure 2c and e, 57 ± 8% of control group value). To determine the effect of WJ-MSCs on an H_2_O_2_-induced fibrosis model, WJ-MSCs (1 × 10^5^ cells) were co-cultured with myotubes treated with H_2_O_2_. WJ-MSC co-cultures showed attenuated fibrosis. Western blotting analysis showed significantly decreased fibronectin (Figure 2d, 78 ± 14% of control group value) and increased MHC (Figure 2e, 73 ± 9% of control group value) levels in myotubes after co-culturing with WJ-MSCs. These data demonstrate that H_2_O_2_ treatment induced fibrosis in myotubes, but that co-culturing with WJ-MSCs can ameliorate this degeneration. To elucidate the mechanisms involved in these changes, we next analyzed the proteins secreted from WJ-MSCs.

### 2.3. Identification of Proteins Secreted from WJ-MSCs in a Co-Culture System

Since WJ-MSCs are known to have a paracrine effect, we investigated proteins secreted from WJ-MSCs in a co-culture system, as described above. After co-culturing WJ-MSCs with myotubes, conditioned media were collected and concentrated 10-fold. A RayBio Biotin Label-Based Human Antibody Array 507 (RayBiotech Life, Peachtree Corners, GA, USA) was used to analyze the levels of secreted proteins present in each of the conditioned media. We found several upregulated proteins in conditioned media of the co-culture group (Figure 3a). Among these, we focused on MMP-1, the protein most strongly upregulated in the co-culture group, as compared to WJ-MSCs alone. MMP-1 is also known to have anti-fibrotic and muscle regeneration effects. To confirm the results of the antibody array, we performed an enzyme-linked immunosorbent assay (ELISA) to measure the levels of human MMP-1 protein in each of the conditioned media. Amounts of human MMP-1 in co-culture media were increased as compared to WJ-MSC only media (Figure 3b, 3965.9 ± 0.8% of control group value).

### 2.4. Human Recombinant MMP-1 Protein Has Protective Effects in an In-Vitro H_2_O_2_-Induced Fibrosis Model

MMP-1 is known to have anti-fibrotic effects, but it has not been confirmed whether MMP-1 is effective in an H_2_O_2_-induced fibrosis model. Therefore, to verify the effect of MMP-1 protein on H_2_O_2_-induced fibrosis, we added human recombinant MMP-1 protein to myotubes treated with H_2_O_2_ and demonstrated that MMP-1 treatment alleviated H_2_O_2_-induced fibrosis (Figure 4a). We also confirmed the alleviating effect of MMP-1 on fibrosis by using Sirius red staining. Collagen deposition was decreased in myotubes and was treated with MMP-1 (Figure 4b, H_2_O_2_: 133 ± 5% of control group value, +MMP-1: 105 ± 8% of control group value). Furthermore, the relative protein expression level of fibronectin was decreased (Figure 4c,d, H_2_O_2_: 149 ± 10% of control group value, +MMP-1: 119 ± 9% of control group value), and that of MHC was increased (Figure 4c,e, H_2_O_2_: 56 ± 4% of control group value, +MMP-1: 80 ± 2% of control group value). Taken together, we confirmed that WJ-MSCs secreted MMP-1 protein in co-culture systems and that this protein has an alleviating effect on H_2_O_2_-induced fibrosis.

### 2.5. MMP-1 Is a Key Factor for the Anti-Fibrotic Effect of Human WJ-MSCs

To determine whether MMP-1 secretion was the major mechanisms involved in the anti-fibrotic effect of WJ-MSCs, MMP inhibitors (tissue inhibitor of metallopeptidase-1 [TIMP-1] and GM6001) were used to treat cells. After treatment with MMP inhibitors, WJ-MSCs were immediately co-cultured with myotubes exposed to H_2_O_2_. As observed above, WJ-MSCs attenuated fibrosis induced by H_2_O_2_ (Figure 5b, H_2_O_2_: 117 ± 3% of control group value, +WJ-MSC: 87 ± 5% of control group value). However, when TIMP-1 or GM6001 was used to treat cells, no anti-fibrotic effects were observed in either group (Figure 5b, +WJ-MSC+TIMP-1: 121 ± 3% of control group value, +WJ-MSC+GM6001: 121 ± 3% of control group value). The relative protein expression level of fibronectin was changed following a similar pattern as seen with Sirius red staining (Figure 5c,d, H_2_O_2_: 183 ± 11% of control group value, +WJ-MSC: 125 ± 7% of control group value, +WJ-MSC+TIMP-1: 225 ± 35% of control group value, +WJ-MSC+GM6001: 177 ± 13% of control group value). In contrast, the expression of MHC was increased in the WJ-MSC co-culture group as compared with the treated group. However, when MMP inhibitors were used, this effect was abrogated (Figure 5c,e, H_2_O_2_: 11 ± 4% of control group value, +WJ-MSC: 37 ± 17% of control group value, +WJ-MSC+TIMP-1: 8 ± 3% of control group value, +WJ-MSC+GM6001: 11 ± 5% of control group value). These results indicated that WJ-MSCs reduced fibrosis caused by oxidative stress mainly via the secretion of MMP-1 protein. 

## 3. Discussion

In most skeletal muscle diseases, including DMD, fibrosis is a major pathogenetic mechanism [2,28,29]. Accumulation of calcium and necrotic fibers is induced by ROS stress, causing fibrosis [30,31,32]. Skeletal muscle fibrosis and accumulation of calcium/necrotic fibers are increased in DMD mice [7,33,34,35]. Several studies have suggested potential anti-fibrotic therapy approaches for DMD [10,36,37]. As WJ-MSCs exert anti-apoptotic effects on skeletal muscle cells [27], we here investigated whether these cells could have anti-fibrotic effects, and also sought to elucidate the main paracrine factor underlying such effects. We showed that intravenous injection of WJ-MSCs into 2–5-month-old C57BL mdx mice significantly decreased fibrosis intensity and accumulation of calcium/necrotic fibers as compared to wild-type mice. We confirmed the effect of WJ-MSCs in an H_2_O_2_-induced fibrosis myotubes model and demonstrated that the paracrine factor MMP-1 is critical for these anti-fibrotic effects. For the first time, we revealed that WJ-MSCs could reduce skeletal muscle fibrosis.

A recent study reported that H_2_O_2_, a ROS with a long half-life, diffuses both within cells and across cell membranes [38,39,40,41]. H_2_O_2_ can damage cellular bio-molecules, such as lipids and protein. It induces oxidative stress and results in chronic fibrosis in DMD patients [6,42,43]. Based on these characteristics, an H_2_O_2_-induced fibrosis model was established as an in-vitro system in this study. We revealed that myotubes exposed to H_2_O_2_ showed typical patterns of fibrosis, including increased expression of fibronectin. Using this in-vitro model, it was possible to study the mechanism underlying the effects of the WJ-MSCs on fibrosis.

A reduction of fibrosis was observed during fibrosis induction in myotubes co-cultured with WJ-MSCs. Furthermore, MMP-1 protein was identified as a paracrine factor through antibody array screening. In order to verify that MMP-1 was the critical factor involved in the anti-fibrotic effects of WJ-MSCs, myotubes were treated with recombinant MMP-1 protein under conditions of oxidative stress. Treatment with recombinant MMP-1 markedly reduced fibrosis in oxidative stress-induced myotubes. We further confirmed that MMP-1 is a key factor by using MMP inhibitor in our in vitro model. Similar to our results, recent studies have reported that MMP-1 has anti-fibrotic effects in a liver fibrosis model and a lacerated skeletal muscle model [44,45,46,47]. Furthermore, the increase in ECM components and upregulation of related genes in DMD patients were reported based on the analysis of differentially expressed gene profile [48]. The amount of secreted protein associated with the reported genes was found to be decreased in the co-culture conditioned media of this study, which showed that our experimental design well reflected the DMD fibrosis model. Our findings demonstrated that human WJ-MSCs play an anti-fibrotic role in skeletal muscle fibrosis via MMP-1.

We also showed increased levels of MHC in this model, but the mechanism for this observation was not identified in this study. Therefore, further study is needed to characterize the effect of MMP-1 treatment on the regenerative function of fibrotic muscle.

DMD is characterized by muscle inflammation and progressive deterioration of muscle mitochondria and function. The absence of dystrophin resulted in excessive calcium penetration into the sarcolemma (the cell membrane). Alterations in calcium and signaling pathways also cause water to enter into the mitochondria, resulting in bursting. Previous studies have documented that MSC-secreted trophic factors highly depend on external environment and status of MSC [49,50]. Some pro-inflammatory cytokines, such as IL-6, TNF-α, can induce MSC transfer of mitochondria to recuse injured cells including retinal cells, cardiomyocytes, and airway epithelia cells [51,52,53]. It appears a pro-inflammatory environment can enhance MSC-mitochondrial transfer and MSC mitochondrial transfer to T cells can in turn educate immune cells (CD4 T cells, i.e.) [54]. In this connection, MSC paracrine functions and mitochondrial transfer capacity are interactive and linked together to promote tissue regeneration. 

It is well known that MSCs of different origins display distinct potential of cell proliferation and survival capacity. For example, MSCs from birth-associated tissues, preferably parts of the placenta and Wharton’s jelly, and pluripotent stem cell-derived MSCs, are highly proliferative and survive longer than adult tissues derived MSCs after transplantation [55,56,57,58].

Taken together, our study suggested that reduction of fibrosis by WJ-MSCs could be a novel treatment strategy for DMD.

## 4. Materials and Methods 

### 4.1. Ethics Statement

This study was approved by the Institutional Animal Care and Use Committee of the Samsung Biomedical Research Institute (SBRI) at Samsung Medical Center. The SBRI is an accredited facility of the Association for Assessment and Accreditation of Laboratory Animal Care International, and abides by the Institute of Laboratory Animal Resources guidelines. In accordance with the guidelines approved by the institutional review board of Samsung Medical Center, umbilical cords were collected with informed consent from pregnant mothers (IRB#2016-07-102).

### 4.2. Cell Culture

Wharton’s jelly-derived mesenchymal stem cells (WJ-MSCs) were cultured according to the standard operating procedures of the Good Manufacturing Practice facility at ENCell Co. Ltd. (Seoul, Korea).

The mouse myoblast cell line, C2C12 (ATCC CRL-1772, American Type Culture Collection, Rockville, MD, USA) was cultured in Dulbecco’s Modified Eagle’s medium (Biowest S.A.S, Nuaille, France) supplemented with 10% fetal bovine serum (Gibco BRL, Carlsbad, CA, USA), 100 U/mL penicillin, and 100 μg/mL streptomycin (Gibco BRL, Carlsbad, CA, USA) in 5% CO_2_ at 37 °C. When myoblasts reached about 80–90% confluence, the culture medium was replaced with a myotube differentiation medium supplemented with 5% horse serum (Gibco BRL, Carlsbad, CA, USA) for 5 days. After differentiation, 2 mM hydrogen peroxide (H_2_O_2_) (Sigma, St. Louis, MO, USA) was added to the growth medium for 24 h. For co-culture experiments, Transwell inserts were placed on the upper side of the myotubes, and WJ-MSCs (1 × 10^5^ cells) were seeded into the Transwell insert (pore size 1 μm, BD Biosciences, Franklin Lakes, NJ, USA) for 24 h under serum-free conditions. The myotubes were treated with or without human recombinant MMP-1 (5 ng/mL) (Sigma, St. Louis, MO, USA) for 24 h in serum-free medium. To determine whether the effect of WJ-MSCs on fibrosis is MMP-1-dependent, human recombinant metallopeptidase inhibitor-1 (TIMP-1) (100 ng/mL) or GM6001 (50 μM) (R&D Systems, Minneapolis, MN, USA), as an MMP inhibitor was used in WJ-MSC co-culture. 

### 4.3. MMP Inhibitors Application

To determine whether the effect of WJ-MSCs on fibrosis is MMP-1-dependent, human recombinant metallopeptidase inhibitor-1 (TIMP-1) (100 ng/mL) or GM6001 (50 μM) (R&D Systems, Minneapolis, MN, USA), as an MMP inhibitor was used. After treatment with H_2_O_2_ for 24 h, WJ-MSCs were co-cultured with or without MMP inhibitors for 24 h in serum-free medium. 

### 4.4. MSCs Administration

Mdx (C57BL/10ScSn-Dmdmdx/J(mdx) (JAX#001801)) and wild-type (C57BL/10ScSnJ (JAX#000476)) mice were purchased from Jackson Laboratory (Bar Harbor, ME, USA), and the offspring of the mice were bred and maintained according to the protocols recommended by the Jackson Laboratory.

Two- to five-month-old mdx mice were placed in a restraining device and injected with a total of 1 × 10^5^ WJ-MSCs (suspended in 100 μL of phosphate buffered saline) through the lateral tail vein. Mice were sacrificed using isoprene 7 days after administration. 

### 4.5. Protein Extraction and Immunoblotting

For total protein extraction, myotubes were scraped from culture dishes and lysed in ice-cold lysis buffer containing RIPA buffer (50 mM Tris-HCl (pH 7.4), 150 mmol/L NaCl, 0.1% sodium dodecyl sulfate (SDS), 1% sodium deoxycholate, 1% NP-40), urea buffer (9.8 mol/l urea, 4% CHAPS, 130 mmol/L dithiothreitol, 40 mmol/L Tris-HCl [pH 8.8], 0.1% SDS), 1 mmol/L ethylenediaminetetraacetate, and a protease/phosphatase inhibitor cocktail. Protein quantification was performed using the Bradford assay (Bio-Rad Laboratories, Hercules, CA, USA). Equivalent amounts of protein were loaded into SDS-polyacrylamide gels, separated, and blotted onto a PVDF membrane (Bio-Rad Laboratories, Hercules, CA, USA). After being blocked with 5% skim milk or bovine serum albumin (BSA), membranes were probed with a specific primary antibody, as follows: anti-fibronectin (ab2413, Abcam, Cambridge, UK), anti-myosin heavy chain (MAB4470, R&D Systems, Minneapolis, MN, USA), or anti-beta actin (sc-47778, Santa Cruz Biotechnology, Dallas, TX, USA). After washing with TBS-T, membranes were incubated with horseradish peroxidase-linked secondary antibody and developed using an enhanced chemiluminescence solution (Bio-Rad Laboratories, Hercules, CA, USA). The band intensities were quantified using Image-Lab software (Bio-Rad Laboratories, Hercules, CA, USA).

### 4.6. Antibody Arrays and Enzyme-Linked Immunosorbent Assay

Proteins secreted in the conditioned media were analyzed using RayBio Biotin Label-Based Human Antibody Array 507 (#AAH-BLG-1-2, RayBiotech Life, Peachtree Corners, GA, USA). All slides were scanned using a GenePix 4100A scanner and analyzed using GenePix Pro 7.0 software (Molecular Devices, San Jose, CA, USA). All data were normalized using internal controls. To determine the concentration of MMP-1 protein secreted in conditioned media, an enzyme-linked immunosorbent assay (#ELH-MMP1-1, RayBiotech Life, Peachtree Corners, GA, USA) was performed according to the recommended protocol. 

### 4.7. Histological Analysis

For staining, myotubes were fixed with 4% paraformaldehyde after co-culturing with WJ-MSCs or treatment with MMP-1 protein. Picro-Sirius red staining (Abcam, Cambridge, UK) was performed to detect collagen deposition in each group, and the image was analyzed using light microscopy, with quantification using Image J (NIH, New Bethesda, MD, USA).

The gastrocnemius muscles of mice were fixed in 4% paraformaldehyde and then embedded in paraffin, sectioned at 4 μm, and stained with Sirius red staining solution or Alizarin red staining solution. Sirius red staining was used to detect fibrosis, and Alizarin red to detect large accumulations of calcium/necrotic fibers, according to standard procedures. Analysis was conducted using a ScanScope AT (Leica Microsystems, Buffalo Grove, IL, USA), and the images were quantified using Image J.

### 4.8. Statistical Analysis

All results were analyzed by *t*-tests or one-way ANOVA followed by Duncan’s multiple comparison using SPSS software (version 23.0; IBM, Armonk, NY, USA). The differences were considered statistically significant at *p* < 0.05, and data are expressed as mean ± SEM.

## Figures and Tables

**Figure 1 ijms-21-06269-f001:**
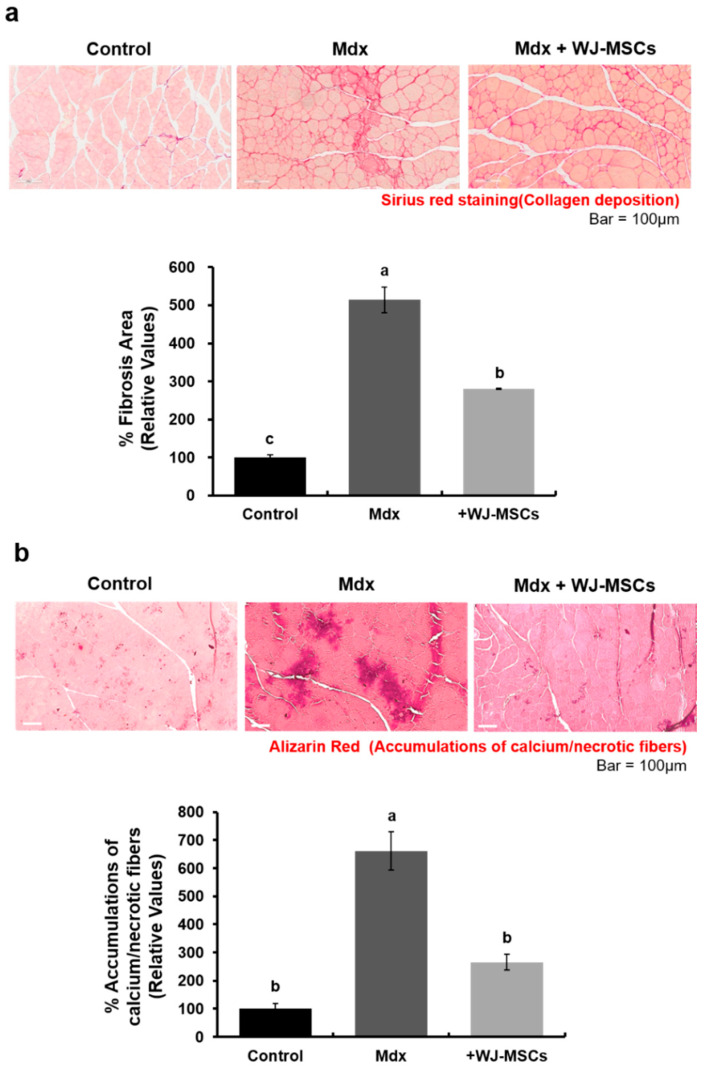
Wharton’s jelly-mesenchymal stem cells (WJ-MSCs) alleviate the fibrosis phenotype in an mdx murine model. (**a**) The relative fibrosis area was increased in the mdx group as compared to the control group. Collagen accumulation was significantly decreased in the WJ-MSC-injection group. Data are expressed mean ± SEM (*n* = 3). Bars with different superscripts are significantly different (one-way ANOVA followed by Duncan multiple range test, *p* < 0.05). (**b**) Alizarin red staining was used to detect intracellular calcium and necrotic fiber levels in skeletal muscle. Intracellular calcium deposition and necrotic fiber accumulation was increased in the mdx group as compared to the control group. However, intravenous injection of WJ-MSCs significantly alleviated this phenotype. Data are expressed as mean ± SEM (*n* = 3). Bars with different superscripts are significantly different (one-way ANOVA followed by Duncan multiple range test, *p* < 0.05).

**Figure 2 ijms-21-06269-f002:**
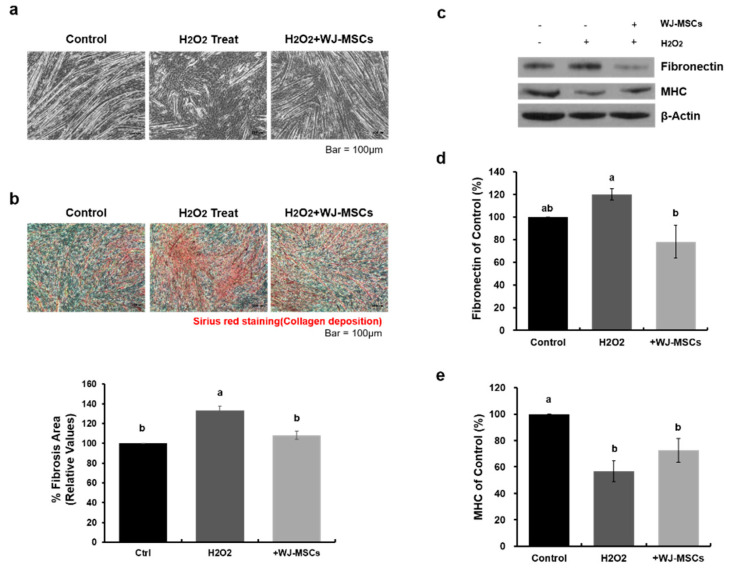
Wharton’s jelly-mesenchymal stem cells (WJ-MSCs) alleviate hydrogen peroxide (H_2_O_2_)-induced fibrosis. (**a**) Myotubes were treated with hydrogen peroxide (H_2_O_2_, 2 mM) to induce a fibrotic phenotype, and were co-cultured with WJ-MSCs for 24 h. Scale bar: 100 μm. (**b**) Representative Sirius red-staining images of myotubes under each condition. A picro-Sirius red staining kit was used to visualize deposition of collagens I and III. Quantitative measurements were made using Image J. Data are expressed mean ± SEM (*n* = 3). Bars with different superscripts are significantly different (one-way ANOVA followed by Duncan multiple range test, *p* < 0.05). Scale bar: 100 μm. (**c**–**e**) Protein levels of fibronectin and myosin heavy chain (MHC) were determined by immunoblotting and were normalized to beta-actin expression. Data are shown as mean ± SEM (*n* = 4). Bars with different superscripts are significantly different (one-way ANOVA followed by Duncan multiple range test, *p* < 0.05).

**Figure 3 ijms-21-06269-f003:**
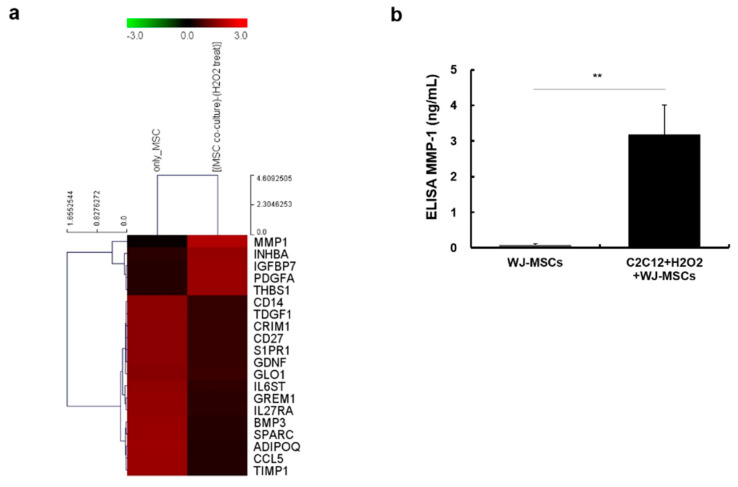
Human antibody array and MMP-1 enzyme-linked immunosorbent assay (ELISA) results in conditioned media. (**a**) Secreted proteins present in the conditioned media were analyzed using the RayBio Biotin Label-Based Human Antibody Array 507. Compared with only Wharton’s jelly-mesenchymal stem cells (WJ-MSCs), the intensity of MMP-1, INHBA, IGFBP7, PDGFA, and THBS1 was increased in co-cultured media. MMP-1: matrix metalloproteinase-1, INHBA: inhibin, beta A, IGFBP7: insulin like growth factor binding protein 7, PDGFA: platelet derived growth factor subunit A, and THBS1: thrombospondin 1. (**b**) Concentration of secreted MMP-1 in each of the conditioned media was measured by using a human MMP-1 ELISA kit. A four-fold increase was observed in the co-cultured media as compared to the WJ-MSC-only media. Data are expressed as mean ± SEM (*n* = 5, ** *p* < 0.01).

**Figure 4 ijms-21-06269-f004:**
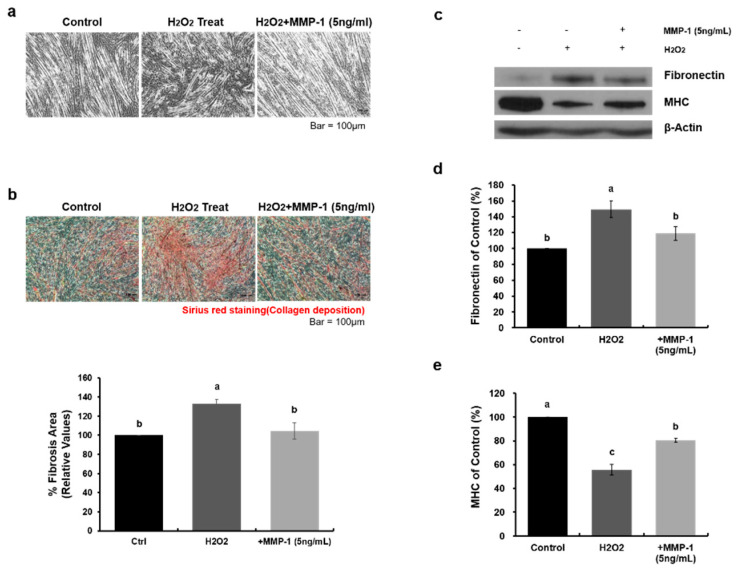
MMP-1 treatment is effective in an in-vitro H_2_O_2_-induced fibrosis model. (**a**) After myotube differentiation, H_2_O_2_ (2 mM) was added to the growth medium, and human recombinant MMP-1 protein (5 ng/mL) was added to the cultures for 24 h. Scale bar: 100 μm. (**b**) The fibrosis level was assessed as a percentage of the area stained with Sirius red staining solution. Quantitative measurements were made using Image J. Data are expressed mean ± SEM (*n* = 3). Bars with different superscripts are significantly different (one-way ANOVA followed by Duncan multiple range test, *p* < 0.05). Scale bars: 100 μm. (**c**‒**e**) Representative images of Western blot analysis to determine protein expression levels of fibronectin and myosin heavy chain (MHC). Beta-actin was used as a loading control. Data are expressed as mean ± SEM (*n* = 3). Bars with different superscripts are significantly different (one-way ANOVA followed by Duncan multiple range test, *p* < 0.05).

**Figure 5 ijms-21-06269-f005:**
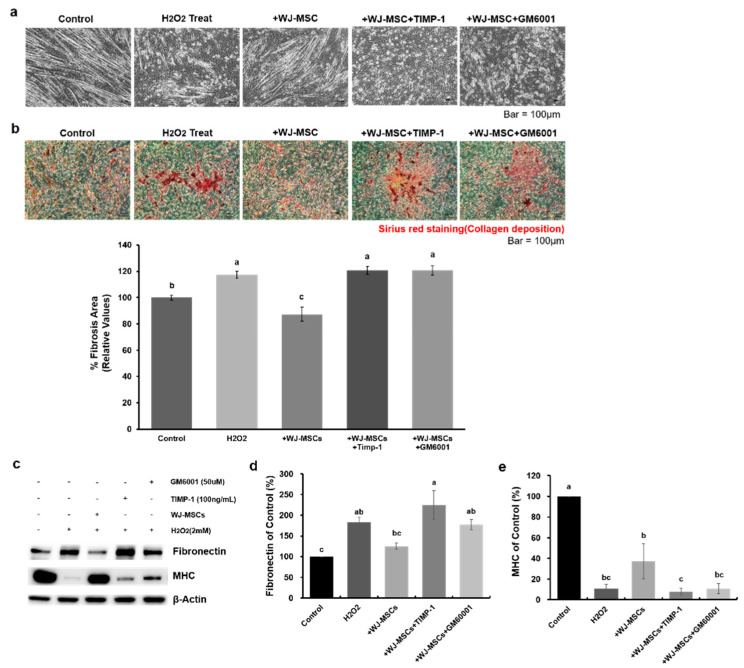
MMP-1 is required for the anti-fibrotic effect of human Wharton’s jelly-mesenchymal stem cells (WJ-MSCs). (**a**) MMP inhibitors (TIMP-1 or GM6001) were used to block the effects of MMP-1 secreted from WJ-MSCs. Fibrosis in myotubes was induced by H_2_O_2_ (2 mM) and WJ-MSCs were co-cultured with the myotubes, with or without MMP inhibitors. Anti-fibrotic effects of WJ-MSCs were not observed when myotubes were treated with MMP inhibitors. Scale bars: 100 μm. TIMP-1: metallopeptidase inhibitor-1. (**b**) Collagen deposition was significantly decreased in the +WJ-MSC group as compared to the treated group. However, these effects were reversed in the MMP inhibitor-treated group. Quantitative measurements were made using Image J. Data are expressed as mean ± SEM (*n* = 3). Bars with different superscripts are significantly different (one-way ANOVA followed by Duncan multiple range test, *p* < 0.05). Scale bars: 100 μm. (**c**–**e**) Relative protein expression level of fibronectin and myosin heavy chain (MHC) was also observed. Fibronectin expression level was significantly decreased in the +WJ-MSC group as compared to the treated group. However, when myotubes were treated with MMP inhibitor, this level was significantly increased. In contrast, relative MHC expression level was increased in the +WJ-MSC group as compared to the Treat treated group, but it was decreased in +WJ-MSC+TIMP-1 and +WJ-MSC+GM6001 group. Beta-actin was used as a loading control. Data are expressed as mean ± SEM (*n* = 3). Bars with different superscripts are significantly different (one-way ANOVA followed by Duncan multiple range test, *p* < 0.05).

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
