# Peer review of "Anti-Fibrotic Effect of Human Wharton’s Jelly-Derived Mesenchymal Stem Cells on Skeletal Muscle Cells, Mediated by Secretion of MMP-1"

_ijms, 2020, doi:10.3390/ijms21176269_

Round 1
Reviewer 1 Report
In this work, Choi A et al., report an in vivo and in vitro anti-fibrotic effect
induced by mesenchymal stem cells derived from Wharton’s jelly either on
dystrophic muscle or in myotubes treated with oxidative stress. Experimental data, strongly suggest that these anti-fibrotic effects are mediated by paracrine secretion of matrix metalloproteinase-1 (MMP-1) from these mesenchymal stem cells.
The results open a research field to identify new potential therapeutic agents
(i.e. human recombinant MMP-1) to treat fibrosis underlying the pathophysiology of Duchenne´s muscular dystrophy among other muscular dystrophies.
The article is well-structured and employed methodology is well applied and described. The conclusions are strong supported.
If the authors consider it appropriate, perhaps the discussion could include a brief correlation of the herein protein profile identified on WJ-MSCs co-culture with the reported gene expression muscle signatures described in human muscular dystrophy patients (i.e. Pescatori et al.FASEB J. 2007 Apr;21(4):1210-26. doi: 10.1096/fj.06-7285com. or Tian et al., Genet Mol Res. 2014 Feb 28;13(1):1402-11. doi: 10.4238/2014.). It could be of utmost interest to determine the viability of MMP-1 as a new anti-fibrotic/therapeutic agent in muscle dystrophies.
Author Response
Reviewer #1 (Remarks to the Author):
In this work, Choi A et al., report an in vivo and in vitro anti-fibrotic effect
induced by mesenchymal stem cells derived from Wharton’s jelly either on
dystrophic muscle or in myotubes treated with oxidative stress. Experimental data, strongly suggest that these anti-fibrotic effects are mediated by paracrine secretion of matrix metalloproteinase-1 (MMP-1) from these mesenchymal stem cells.
The results open a research field to identify new potential therapeutic agents
(i.e. human recombinant MMP-1) to treat fibrosis underlying the pathophysiology of Duchenne´s muscular dystrophy among other muscular dystrophies.
The article is well-structured and employed methodology is well applied and described. The conclusions are strong supported.
Reply: We appreciate the reviewer’s meticulous review of our manuscript.
If the authors consider it appropriate, perhaps the discussion could include a brief correlation of the herein protein profile identified on WJ-MSCs co-culture with the reported gene expression muscle signatures described in human muscular dystrophy patients (i.e. Pescatori et al.FASEB J. 2007 Apr;21(4):1210-26. doi: 10.1096/fj.06-7285com. or Tian et al., Genet Mol Res. 2014 Feb 28;13(1):1402-11. doi: 10.4238/2014.). It could be of utmost interest to determine the viability of MMP-1 as a new anti-fibrotic/therapeutic agent in muscle dystrophies.
Reply: As the reviewer has suggested, we have confirmed the references mentioned above, and we have discussed about the issues with proper reference (Pescatori et al.FASEB J. 2007 Apr;21(4):1210-26).(line 259, Ref 48)
Reviewer 2 Report
The elegant study is performed. The study design is appropriate to answer the aim.
It is demonstrated in the Article that human WJ-MSCs play an anti-fibrotic role in skeletal muscle fibrosis via MMP-1 and therefore could be a promising novel treatment strategy for DMD. Additional research question for further study is also formulated.
Probably it would be better to add the examples of hypotheses about the cause of fibrosis in DMD to underline that reactive oxygen species (ROS)-induced oxidative stress is a major cause of this feature of DMD (Lines 92-94)?
It is indicated in the Lines 198, 199: Anti-fibrotic effects of WJ-MSCs were not observed when myotubes were treated with MMP inhibitors.
And in Lines 267, 268 it is indicated: MMP inhibitor was used in WJ-MSC co-culture.
Probably it would be better to specify the methodology of MMP inhibitors application?
It would be probably better to add an explanation of myosin heavy chain (MHC) level changes in different groups to the Figure 5 description (Line 205)?
Should be the number of pages indicated in the following reference?:
8. Forcina, L.; Pelosi, L.; Miano, C.; Musarò, A. Insights into the Pathogenic Secondary Symptoms Caused 349 by the Primary Loss of Dystrophin. Journal of Functional Morphology and Kinesiology 2017, 2, doi:10.3390/jfmk2040044. (Line 349)
Author Response
Reviewer #2 (Remarks to the Author):
The elegant study is performed. The study design is appropriate to answer the aim.
It is demonstrated in the Article that human WJ-MSCs play an anti-fibrotic role in skeletal muscle fibrosis via MMP-1 and therefore could be a promising novel treatment strategy for DMD. Additional research question for further study is also formulated.
Reply: We appreciate the reviewer for providing us with positive feedback and for carefully reviewing our paper.
Probably it would be better to add the examples of hypotheses about the cause of fibrosis in DMD to underline that reactive oxygen species (ROS)-induced oxidative stress is a major cause of this feature of DMD (Lines 92-94)?
Reply: We appreciate the reviewer for addressing an important issue. We have included appropriate examples related with cause of fibrosis in DMD.
In response to the reviewer’s comments, we added this issue to revised manuscript as follows.
In Results
“To investigate the role of WJ-MSCs on the fibrosis observed in mdx mice, we established an in-vitro skeletal muscle fibrosis model. There are many possible hypotheses about the cause of fibrosis in DMD, such as aging, trauma, or inflammation [4], but reactive oxygen species (ROS)-induced oxidative stress is considered as a major cause of this feature of DMD.” (line 96-98)
It is indicated in the Lines 198, 199: Anti-fibrotic effects of WJ-MSCs were not observed when myotubes were treated with MMP inhibitors.
And in Lines 267, 268 it is indicated: MMP inhibitor was used in WJ-MSC co-culture.
Probably it would be better to specify the methodology of MMP inhibitors application?
Reply: We appreciate the reviewer for addressing an important issue.
In response to the reviewer’s comments, we added this issue to revised manuscript as follows.
In Materials and methods (line 309-313)
“4.3. MMP inhibitors application
To determine whether the effect of WJ-MSCs on fibrosis is MMP-1-dependent, human recombinant metallopeptidase inhibitor-1 (TIMP-1) (100 ng/mL) or GM6001 (50 μM) (R&D Systems, Minneapolis, MN), as an MMP inhibitor was used. After treatment with H2O2 for 24 hours, WJ-MSCs were co-cultured with or without MMP inhibitors for 24 hours in serum-free medium.
It would be probably better to add an explanation of myosin heavy chain (MHC) level changes in different groups to the Figure 5 description (Line 205)?
Reply: According to the reviewer’s suggestion, we have included an explanation of the change of MHC level in Figure 5 description. (line 221-228)
In Figure 5 legends
c‒e. Relative protein expression level of fibronectin and myosin heavy chain (MHC) was also observed. Fibronectin expression level was significantly decreased in the +WJ-MSC group as compared to the Ttreated group. However, when myotubes were treated with MMP inhibitor, this level was significantly increased. In contrast, relative MHC expression level was increased in the +WJ-MSC group as compared to the Treat treated group, but it was decreased in +WJ-MSC+TIMP-1 and +WJ-MSC+GM6001 group. Beta-actin was used as a loading control. Data are expressed as mean ± SEM (n = 3, *p < 0.05). Bars with different superscripts are significantly different (one-way ANOVA followed by Duncan multiple range test, p < 0.05).
Should be the number of pages indicated in the following reference?
- Forcina, L.; Pelosi, L.; Miano, C.; Musarò, A. Insights into the Pathogenic Secondary Symptoms Caused 349 by the Primary Loss of Dystrophin. Journal of Functional Morphology and Kinesiology 2017, 2, doi:10.3390/jfmk2040044. (Line 349)
Reply: In response to the reviewer’s comments, we have added the number of pages in the mentioned reference. (line 394-396)
Reviewer 3 Report
This manuscript presented an interesting study on the anti-fibrotic effect of human Wharton's Jelly-derived Mesenchymal Stem Cells (WJ)-MSCs. Based on the mdx animal model and hydrogen peroxide-induced fibrosis model in vitro, the short-term anti-fibrosis effect of WJ-MSCs was confirmed. At the same time, MMP-1 was identified to be a critical paracrine factor in the anti-fibrotic effects. Thus, they concluded that this work proposed a novel target for treatment of fibrosis. that WJ-MSCs exert anti-fibrotic effects against skeletal muscle fibrosis, primarily via MMP-1, indicating a novel target for treatment in muscle diseases, such as DMD.
Overall, this article provides useful information for researchers and clinicians to understand biological roles of mesenchymal stem cells for the Duchenne muscular dystrophy (DMD). However, there are several issues need to be addressed for above conclusion .
Major concerns
- The method of statistical analysis in the part of M&M is problematic . In the results , the authors performed comparisons more than two groups in many figures, it is inappropriate to employ student’s t test . Help from statistical expert is required if authors are not aware how to use appropriate analysis tools .
- The limitation is this study should be addressed in discussion. Duchenne muscular dystrophy (DMD) is characterized by the muscle inflammation and progressive deterioration of muscle mitochondria and function.The absence of dystrophin resulted in excessive calcium penetration into the sarcolemma (the cell membrane). Alterations in calcium and signalling pathways also cause water to enter into the mitochondria, which then burst. The authors described possible mechanisms of WJ -MSC on protection of DMD from fibrosis is through paracrine factors including MMP-1 secreted from However, the capacity and contents of MSC-secreted MMP-1 are changeable which is highly depending on status of MSC, either at rest status or stimulated status ( i.e. pro-inflammatory stimuli). Previous studies have documented that MSC-secreted trophic factors highly depends on external environment and status of MSC (Expert Rev Cardiovasc Ther. 2013 Apr;11(4):505-17; Thromb Haemost 2010; 104(01): 6-12;). Some pro-inflammatory cytokines i.e. IL-6,TNF-a can induce MSC transfer of mitochondria to recuse injured cells including retinal cells, cardiomyocytes and airway epithelia cells ( i.e. Theranostics. 2019; 9(8): 2395–2410. Stem cell reports 11 (5), 1120-1135, 2018.; Stem Cell Reports. 2016 Oct 11;7(4):749-763.). It appears a pro-inflammatory environment can enhance MSC- mitochondrial transfer and MSC mitochondrial transfer to T cells in turn can in turn educate immune cells ( CD4 T cells i.e. EMBO Rep. 2020 Feb 5;21(2):e48052) . In this connection, MSC paracrine functions and mitochondrial transfer capacity are interactive and linked together to promote tissue regeneration. It will be valuable to include these contents and references in discussion for more insights into understanding multiple facets of MSC-modulated tissue regeneration including DMD.
- What potential of proliferation and survival capacity of WJ-MSC post transplantation It is well known that different origins -derived MSCs display different potential of cell proliferation and survival capacity . For examples, pluripotent stem cell-derived MSCs are highly proliferative and survive longer than bone marrow MSCs after transplantation ( Am J Physiol Cell Physiol. 2012 Jul 15;303(2):C115-25.; 2010;121:1113–1123) . It will be informative to include these contents in discussion and will be helpful to choose right MSC in appropriate applications.
Specific comments:
- Line 48~49, ‘There are several approaches to resolving fibrosis in DMD patients, but no effective treatment has been developed to date’, and Line214-215, ‘Several studies have suggested potential anti-fibrotic therapy approaches for DMD’. In the introduction section and discussion section, the authors mentioned several approaches to resolve fibrosis which were thought to be important background information. What these approaches are? Please provide more elaborations.
- Line 61, ‘XCL1 protein’. What is the abbreviation for ‘XCL1’?
- Line 166. In figure caption ‘Figure 4: MMP-1 treatment is effective in an in-vitro H2O2-induced fibrosis model’, H2O2 were clerical error.
Author Response
Reviewer #3 (Remarks to the Author):
This manuscript presented an interesting study on the anti-fibrotic effect of human Wharton's Jelly-derived Mesenchymal Stem Cells (WJ)-MSCs. Based on the mdx animal model and hydrogen peroxide-induced fibrosis model in vitro, the short-term anti-fibrosis effect of WJ-MSCs was confirmed. At the same time, MMP-1 was identified to be a critical paracrine factor in the anti-fibrotic effects. Thus, they concluded that this work proposed a novel target for treatment of fibrosis. that WJ-MSCs exert anti-fibrotic effects against skeletal muscle fibrosis, primarily via MMP-1, indicating a novel target for treatment in muscle diseases, such as DMD.
Overall, this article provides useful information for researchers and clinicians to understand biological roles of mesenchymal stem cells for the Duchenne muscular dystrophy (DMD). However, there are several issues need to be addressed for above conclusion.
Reply: We would like to appreciate the reviewer for providing us with positive feedback.
Major concerns
- The method of statistical analysis in the part of M&M is problematic. In the results, the authors performed comparisons more than two groups in many figures, it is inappropriate to employ student’s t test. Help from statistical expert is required if authors are not aware how to use appropriate analysis tools.
Reply: We appreciate the reviewer for addressing an critical issue. As recommended by the reviewer, One-way ANOVA (analysis of variance) was performed, and then Duncan’s multiple comparison test was performed for a comparison of two or more groups.
According to the reviewer’s suggestion, we have included an explanation of the Statistical analysis in Figure 1,2,4, and 5 description
The limitation is this study should be addressed in discussion. Duchenne muscular dystrophy (DMD) is characterized by the muscle inflammation and progressive deterioration of muscle mitochondria and function. The absence of dystrophin resulted in excessive calcium penetration into the sarcolemma (the cell membrane). Alterations in calcium and signalling pathways also cause water to enter into the mitochondria, which then burst. The authors described possible mechanisms of WJ -MSC on protection of DMD from fibrosis is through paracrine factors including MMP-1 secreted from However, the capacity and contents of MSC-secreted MMP-1 are changeable which is highly depending on status of MSC, either at rest status or stimulated status ( i.e. pro-inflammatory stimuli). Previous studies have documented that MSC-secreted trophic factors highly depends on external environment and status of MSC (Expert Rev Cardiovasc Ther. 2013 Apr;11(4):505-17; Thromb Haemost 2010; 104(01): 6-12;). Some pro-inflammatory cytokines i.e. IL-6, TNF-a can induce MSC transfer of mitochondria to recuse injured cells including retinal cells, cardiomyocytes and airway epithelia cells ( i.e. Theranostics. 2019; 9(8): 2395–2410. Stem cell reports 11 (5), 1120-1135, 2018.; Stem Cell Reports. 2016 Oct 11;7(4):749-763.). It appears a pro-inflammatory environment can enhance MSC- mitochondrial transfer and MSC mitochondrial transfer to T cells in turn can in turn educate immune cells (CD4 T cells i.e. EMBO Rep. 2020 Feb 5;21(2):e48052). In this connection, MSC paracrine functions and mitochondrial transfer capacity are interactive and linked together to promote tissue regeneration. It will be valuable to include these contents and references in discussion for more insights into understanding multiple facets of MSC-modulated tissue regeneration including DMD.
Reply: We appreciate the reviewer for addressing an important issue. We believe that the reviewer has raised an important issue and agree with the reviewer’s comment. In response to the reviewer’s comments, we added this issue to revised manuscript as follows.
In Discussion (line 266-276)
DMD is characterized by muscle inflammation and progressive deterioration of muscle mitochondria and function. The absence of dystrophin resulted in excessive calcium penetration into the sarcolemma (the cell membrane). Alterations in calcium and signaling pathways also cause water to enter into the mitochondria, resulting in bursting. Previous studies have documented that MSC-secreted trophic factors highly depend on external environment and status of MSC [49,50]. Some pro-inflammatory cytokines, such as IL-6, TNF-α, can induce MSC transfer of mitochondria to recuse injured cells including retinal cells, cardiomyocytes, and airway epithelia cells [51-53]. It appears a pro-inflammatory environment can enhance MSC- mitochondrial transfer and MSC mitochondrial transfer to T cells in turn can in turn educate immune cells (CD4 T cells i.e) [54]. In this connection, MSC paracrine functions and mitochondrial transfer capacity are interactive and linked together to promote tissue regeneration.
What potential of proliferation and survival capacity of WJ-MSC post transplantation It is well known that different origins -derived MSCs display different potential of cell proliferation and survival capacity. For examples, pluripotent stem cell-derived MSCs are highly proliferative and survive longer than bone marrow MSCs after transplantation (Am J Physiol Cell Physiol. 2012 Jul 15;303(2):C115-25.; 2010;121:1113–1123). It will be informative to include these contents in discussion and will be helpful to choose right MSC in appropriate applications.
Reply: We appreciate the reviewer for addressing an important issue. We believe that the reviewer has raised an important issue and agree with the reviewer’s comment.
It has been reported that compared to adult MSCs, WJ-MSCs showed the highest proliferation activity (Batsali, A. K. et al., 2017; Baksh, D et al., 2007; Lu, L. L. et al., 2006). Also, WJ-MSCs have several advantages over adult MSCs in general. They are easily isolated from umbilical cord(UC) which is readily available; the UC is considered as a medical waste and is discarded at birth. Thus, unlike BM-MSCs which require painful BM-aspiration, the isolation of WJ-MSCs is non-invasive. Moreover, several reports showed a relatively high expression of pluripotency markers in WJ-MSCs compared to MSCs from other sources, implying a more primitive status (Fong et al., 2011; El Omar et al., 2014). El Omar et al., 2014 showed that WJ-MSCs exhibit a unique gene expression profile compared to BM-MSCs using the high throughput single-cell RNA-sequencing technique. In this report, 436 genes were found to be significantly differentially expressed when comparing the two cell types. Those genes are related to several cellular processes such as chemotaxis, apoptosis, anti-tumor activity, and immuno-modulation. Indeed, the BM-MSC senescence is earlier than the WJ-MSC senescence (Batsali AK et al., 2017). Those differences might at least in part explain many of the advantages which WJ-MSCs have over BM-MSCs.
In response to the reviewer’s comments, we added this issue to revised manuscript as follows.
In Discussion (line 277-280)
It is well known that MSCs of different origins display distinct potential of cell proliferation and survival capacity. For examples, MSCs from birth-associated tissues, preferably parts of the placenta and Wharton's jelly, and pluripotent stem cell-derived MSCs are highly proliferative and survive longer than adult tissues derived MSCs after transplantation [55-58]